# Regulatory Effects of Alhagi Honey Small-Molecule Sugars on Growth Performance and Intestinal Microbiota of Lambs

**DOI:** 10.3390/ani14162402

**Published:** 2024-08-19

**Authors:** Jianlong Li, Tuerhong Kudereti, Adelijiang Wusiman, Saifuding Abula, Xiaodong He, Jiaxin Li, Yang Yang, Qianru Guo, Qingyong Guo

**Affiliations:** 1College of Veterinary Medicine, Xinjiang Agricultural University, Urumqi 830052, China; pbb0108@163.com (J.L.); tur2873862732@163.com (T.K.); adeli5621231@163.com (A.W.); saifudingabula@163.com (S.A.); shouyihexiaodong@sina.com (X.H.); 13999249170@163.com (J.L.); sdwfyy9809@163.com (Y.Y.); ph18299335758@163.com (Q.G.); 2Xinjiang Key Laboratory of New Drug Study and Creation for Herbivorous Animals, Urumqi 830052, China

**Keywords:** Alhagi honey small molecule sugar, lamb, growth performance, immune function, intestinal microbiota

## Abstract

**Simple Summary:**

Plant sugars have a good effect on promoting animal body development and immune function. This study confirmed that AHAS feeding in unweaned lambs can effectively promote the growth performance of lambs, enhance antioxidant activity, promote the secretion of antibodies and cytokines, increase the abundance of intestinal beneficial bacteria, reduce the number of harmful bacteria, and effectively promote the secretion of intestinal short-chain fatty acids.

**Abstract:**

The present study was designed to assess the impact of Alhagi honey small-molecule sugars (AHAS) on Hu lambs. Therefore, in this study, AHAS low-dose (AHAS-L, 200 mg/ kg per day), AHAS medium-dose (AHAS-M, 400 mg/kg per day), and AHAS high-dose (AHAS-H, 800 mg/kg per day) were administered to Hu lambs to investigate the regulatory effects of AHAS on growth performance, oxidation index, immune system enhancement, and intestinal microbiota. The results showed that lambs in the AHAS-H group exhibited significantly increased in average daily weight gain, and growth performance compared to those in the control group (*p* < 0.05). Moreover, AHAS-H supplementation resulted in increased levels of serum antioxidant enzymes (SOD, GSH-Px, and T-AOC), serum antibodies (IgA, IgG, and IgM), and cytokines (IL-4, 10,17, IFN-γ, and TNF-α) compared with the control group (*p* < 0.05). Additionally, it increased the quantity and richness of beneficial bacteria at such as *Sphingomonas*, *Ralstonia*, and *Flavobacterium*, activating various metabolic pathways and promoting the production of various short-chain fatty acids. In summary, our findings highlight the potential of AHAS-H treatment in enhancing intestinal health of lambs by improving intestinal function, immunity, and related metabolic pathways. Consequently, these results suggest that AHAS holds promising potential as a valuable intervention for optimizing growth performance and intestinal health in lambs.

## 1. Introduction

Recently, there has been much research on a variety of dietary supplements, such as prebiotics, vitamins, and plant-derived supplements, to determine their effectiveness in regulating gut bacteria and promoting digestive health [1]. Among these, sugars obtained from the different plants used in intensive farming have been assessed for enhancing gastrointestinal health, immunity, and growth performances [1]. In recent years, China’s lamb production has shifted towards large-scale monoculture-based industrial farming in response to increasing consumer demand for meat. In this context, sheep husbandry often involves a large number of antibiotics or chemical immune stimulants in-feed to protect the fragile health of lambs [2]. Conversely, overuse of these medications has resulted in additional adverse effects. An instance of antibiotic misuse in agriculture in the presence of residual drugs, disrupted the microbacterial ecology and contributed to the development of antibiotic resistance [3]. Different studies have reported that plant-derived supplements, particularly plant sugars, enhance growth performance, immunological responses, antioxidant range, and alteration of gut flora in lambs [4]. According to Wang et al., the inclusion of wheat bran ferrolyl oligosaccharides (FOs) in lambs’ diets resulted in enhanced growth performance and increased levels of glutathione peroxidase (GSH-Px), CAT (catalase), SOD (superoxide dismutase), and antioxidative range (T-AOC) levels [5]. Chen et al. demonstrated a substantial improvement in the average weight gain of lambs, as well as increase in the serum IgA, IgM and IgG levels, when Chinese medicine polysaccharides (CMPs) were added to lambs’ diets [6]. Guo et al. discovered that when lambs were administered a gavage of fucoidan, their intestinal microbiota, including *Bacteroidetes* and *Actinobacteria* together with short-chain fatty-acids (SCFAs) including propionate and butyrate, increased [7]. Experiment demonstrated that administering fructo-oligosaccharides to lambs showed a substantial improvement in average daily weight and promoted the increased occurrence of *Firmicutes*, *Lactobacillus*, *Bacteroides*, and *Actinobacteria* [8]. 

Alhagi honey is a sweetener that is obtained from the fluids expelled by *Alhagi pseudalhagi*. Alhagi honey, also referred to as “thorny honey” or “acacia honey”, has been used in China for over two millennia as a natural remedy [9]. Alhagi honey is recorded in such books as the “Compendium of Materia-Medica”, “Supplementary Materia Medica”, and the “History of the North” [10]. Alhagi honey is mostly sourced from Xinjiang province in northwest China, as well as desert Central Asia and other nearby regions such as Pakistan, Afghanistan, northern India, Iran, Iraq, Kazakhstan, Syria, and Mongolia [11]. Modern pharmacological research has shown that Alhagi honey has a combination of short- and large-molecule sugars as its main components [12,13]. Cai et al. discovered that Alhagi honey sugars had a substantial impact on the average daily growth and levels of IL-4 and 17, and IFN-γ in the intestines of chickens. Furthermore, it was shown that they increase the abundances of *Proteobacteria*, *Bacteroidetes*, and *Firmicutes*, and the concentration of SCFAs like acetic acid, propionic acid, and butyric acid in broiler chickens [14]. Administration of Alhagi honey polysaccharides to mice significantly enhanced the release of cytokines (IL-4, IL-10, IL 17, TNF-α, and IFN-γ) and enhanced the richness of the intestinal microflora (*Firmicutes*, *Prevotella*, *Lactobacillus*, and *Bacteroidetes*) [15,16]. But currently Alhagi honey sugars’ influence on growth performance, immunological function, SCFAs, and gut microbiota composition in lambs remains uncertain. 

Therefore, the main sugars from Alhagi honey were isolated and the functional groups and monosaccharides of Alhagi honey small-molecule sugars (AHAS) were analyzed by spectroscopic and chromatographic methods. The AHAS was orally administered to lambs and the effects of AHAS on growth performance, antioxidant levels, antibody and cytokine levels, intestinal microbes and SCFAs were evaluated. This research aims to evaluate the effects of AHAS on growth promotion, immune enhancement, and gastrointestinal health of lambs, and thus provide alternative antibiotic materials that support the healthy growth of lambs.

## 2. Materials and Methods

### 2.1. AHAS Extraction

The small chemicals included in Alhagi honey were eliminated by subjecting it to reflux with petroleum ether (ratio 1:10 *w*/*v*). The liquid component was extracted with distilled water and the extract was placed in a dialysis bag (3000 Da); the macromolecular polysaccharides (polysaccharides with a molecular weight above 3000 Da) in the dialysis bag were discarded; and the part outside the dialysis bag was collected and concentrated to obtain a concentrated form of Alhagi honey small-molecule sugars (AHAS, this part consists of monosaccharides and oligosaccharides with molecular weights between 200 and 3000 Da). The AHAS accounts for 76.56% of the total Alhagi honey sugar content. 

### 2.2. Component Separation of AHAS

To isolate the different sugar components in the AHAS, we utilized a diethylaminoethyl cellulose-650M (DEAE-650M) column with dimensions of 2.6 cm × 50 cm and a flow rate of 1 mL/min. This column was obtained from BestChrom Biosciences in Shanghai, China (product code AI0034). The main components in AHAS were ultimately produced by the process of freeze-drying. 

### 2.3. Structural Evaluation of AHAS

#### 2.3.1. Assessment of Monosaccharide Content

The monosaccharide content of AHAS was assessed by using the technique of anion-exchange chromatography (AEC) at the high-performance setting [17]. An amount of 5 mg of AHAS was subjected to hydrolysis using 2 molar (M) trifluoro-acetic acid in a sealed container. After being subjected to three methanol rinses, the hydrolyzed sample was then mixed well in deionized water (DW). The solution underwent filtration and analysis utilizing a Dionex™-CarboPac™-PA20 column (Dionex, 3 × 1 50 mm) coupled with an ICS 5000 Thermo-Fisher Scientific chromatography system in the Waltham, MA, USA. 

#### 2.3.2. Spectral Valuation by Fourier Transforms Infrared (FT-IR)

The AHAS structure was established by spectroscopy using a Fourier Transform Infrared (FT-IR) method [18]. A dried form of potassium bromide (KBr) was used to roughly pulverize an amount of 1 milligram of AHAS before applying pressure to compact it onto a tiny plate. The infrared spectra (IS) within the 4000 to 400 cm^−1^ wavelength range was then analyzed using the spread spectra Fourier-transform-infrared (FT-IR) spectrometer.

### 2.4. Animal Experimental Design

The experiment was conducted at Puhui Township farm in Korla City. The experiment was designed after the approval of the Ethical Committee (EC) of Animal Welfare (EC approval no. 2022016) at Xinjiang Agricultural University to ensure compliance with all the ethical laws of regulation and the relevance of testing and research conducted on animals. A total of 28 Hu lambs (female lambs), with each lamb weighing 2.13 ± 0.12 kg and being around 7 days old, were randomly divided into 4 groups (*n* = 7): the control group with the administration of an equal volume of normal saline daily, and the treatment groups, which were divided into the AHAS low-dose group (AHAS-L, 200 mg/kg per day), AHAS medium-dose group (AHAS-M, 400 mg/kg per day), and AHAS high-dose group (AHAS-H, 800 mg/kg per day). Doses were administered for 28 days. Lambs were weighed weekly and drug concentrations were increased according to weight gain. During the whole study trial, ewes and lambs were housed and managed in the same group under the same environmental conditions. Feed and water levels for each ewe were uniformly managed throughout the experiment period, and lambs were allowed to feed freely from their mothers. Specifically, lambs stayed with ewes for 17 days, feeding from lactation ad libitum. After 17 days, the lambs were allowed to drink water ad libitum and fed lamb pellet feed (Zhengbang Feed Co., LTD, Tianjin, China), but lactation was limited to 10 h per day.

### 2.5. Effect of AHAS on Lambs’ Growth Performance

In the whole study trial, the clinical and physical conditions of each group were recorded and observed on a daily basis. In addition, the weight and body size of each lamb was measured every week. At the start and end of the study trial, all the measurements were taken, such as body weight, height, slope length, bust length, and pipe length circumference. 

### 2.6. Effect of AHAS on Levels of Immunoglobulin, Cytokines, and Oxidation Indices

After the 28th day of the study, blood samples were collected in 10 mL vacutainer vials before morning feeding. The collected blood samples were analyzed by centrifugation at 3000 rpm for 15 min at 25 °C (TDL-40B, Anting Scientific Instrument Factory, Shanghai, China). After the centrifugation process, serum was collected carefully for further analysis and stored at −80 ℃.

Oxidation indices: T-AOC, SOD, GSH-Px, CAT, and MDA concentrations were recorded using the ELISA kit (Nanjing Jiengcheng Biotechnology Co., Ltd., Nanjing, China). The total concentrations of IgM, IgG, and IgA were recorded, as were the concentrations of TNF-α, IFN-γ, and IL-4, 10, and 17, with the help of an ELISA kit (FANKEW, Shanghai Kexing Trading Co., Ltd., Shanghai, China). The manufacturer’s protocol method was also followed at the time of analysis [19].

### 2.7. Analysis of 16S rRNA Sequencing and Extraction of Fecal DNA

After study trial completion, the fecal samples were collected from each group of lambs (*n* = 7, per group) and the samples were put in dry ice and sent to the laboratory. The sequencing technique of 16S rRNA was used to assess the treatment of the samples from the control and AHAS-H groups. The total genomic DNA from stool samples was isolated using a E.Z.N.A.^®^ Soil DNA Kit (OMEGA bio-tek Innovations in nucleic acid isolation, Atlanta, GA, USA). PCR amplification was conducted by the analysis of 16S rRNA sequences using common (universal) primers 515F (5′-GCACCTATGGGCTTAAAGNG-CAG-3′) and 805R (5′TACNAGGGTATCTAATCC-3′). Next, the testing of the amplicons was evaluated for their purity and integrity. The sequence identified library was created using the MiSeq Reagent-Kit v3 from Illumina, USA. Then, the library was accurately quantified by Qubit. The amplicon library was sequenced by Genesky Biotechnology Inc. (Shanghai, China) using an Illumina MiSeq Benchtop Sequencer (Illumina, San Diego, CA, USA) with an Ilumina 2 × 250 bp double-end sequencing strategy [16].

### 2.8. SCFAs Concentrations Determined in the Fecal Content

The samples were obtained by using lamb feces (0.5 g of fecal stool, *n* = 7 per group). The gas chromatograph (GC) from Thermo-Fisher Scientific, USA was used to quantify the content of SCFAs in the lamb feces [20].

### 2.9. Statistical Evaluation/Analysis

The statistical evaluation of the study used different software and tools: SPSS 24.0 to perform analysis of variance, and one-way ANOVA for observation of the differences. The graphical explanation was performed using GraphPad Prism 9. All the outcomes of this study were represented by the mean ± SEM.

## 3. Results

### 3.1. AHAS Primary Structure

#### 3.1.1. AHAS Isolation and Monosaccharide Composition

The AHAS samples, after being separated from the gel fiber DEAE-650M, showed two distinct accumulation peaks in the emission curves as shown in Figure 1A, named AHAS-1 and AHAS-2. The most significant aggregation peak between tubes 15 to 60 occurred after extraction of AHAS-1 from deionized water, with 97.98% of the total AHAS sugar content.

The monosaccharide composition of the polysaccharide components detected by HPLC is shown in Figure 1B. AHAS is primarily formed of mannose (Man), arabinose (Ara), rhamnose (Rha), xylopentaose (Xyl), glucose (Glc), and galactose (Gal), with molar (M) ratio of 0.77:1.8:1:3.79:1.1:73.92. Gal (β-D-Galactopyranosyl ester) is the main monosaccharide backbone of AHAS.

#### 3.1.2. Infrared Spectra of the AHAS

The infrared spectral results of AHAS, presented in Figure 2, show broad absorption peaks at 3277.27 cm^−1^ for AHAS. This peak is attributed to the -OH enlarging vibration absorption of polysaccharides, representing the presence of intramolecular or intermolecular hydrogen bonding, which broadens the absorption peaks due to hydrogen bonding by hydroxyl groups [21]. Additionally, the weak absorption peaks at 2929.94 cm^−1^ can be assigned to the enlarged beat of the C-H bond of alkyl groups in polysaccharides [22]. This suggests that AHAS is mainly composed of polysaccharides, which usually contain hydroxyl (-OH) and alkyl groups. The strong absorption peaks at 1641.77 cm^−1^ and 1413.90 cm^−1^ correspond to the C=O bending beat and the C-H bending beat, respectively [21]. The absorption peak at 1263.37 cm^−1^ is attributed to the C-H enlarging vibration, while the peaks at 1103.34 cm^−1^ and 1039.09 cm^−1^ result from the C-O-C enlarging beat in the pyranose ring, which indicates the presence of pyranose in AHAS [22]. Furthermore, the signals at 987.30 cm^−1^, 923.25 cm^−1^, and 832.72 cm^−1^ are revealing of D-glucose, α-D-glucan, and D-glucopyranose derivatives, respectively [21,22].

### 3.2. Effect of AHAS on Average Weight and Daily Gain of Lambs

Table 1 presents a concise overview of the growth and progress of lambs. None of the experimental groups exhibited any statistically significant differences in initial body weight, height, slope length, bust length, or pipe length circumference compared with the control group (*p* > 0.05). Supplementing with AHAS had a notable impact on both body weight and daily growth over the 28-day feeding period. Table 1 demonstrates that lambs in the AHAS-H group had a remarkably higher body weight and average daily growth with the control group (*p* < 0.05). The increase was directly proportional to the dose. The control group exhibited considerably smaller final body height, initial body slope length, and final bust length in comparison to the AHAS-H groups (*p* < 0.05), with a gradual rise in relation to the dosage. It may be inferred that AHAS effectively enhanced the growth and skeletal development of lambs.

### 3.3. Effect of AHAS on the Antioxidant Status of Lambs

The effects of AHAS-L, AHAS-M, and AHAS-H on antioxidant rank in lambs are presented in Figure 3. The serum T-AOC, T-SOD, and GSH-PX activity was markedly improved in the AHAS-H group compared with the control group (*p* < 0.05). Moreover, the AHAS-H groups exhibited significantly lower serum MDA activity than the control (*p* < 0.05).

### 3.4. AHAS Effect on Lamb Serum Immunoglobulins

Figure 4 demonstrates that the group receiving AHAS-H supplementation had markedly raised levels of plasma IgA, IgG, and IgM concentrations compared with the control group (*p* < 0.05). Furthermore, the groups that received AHAS-H and AHAS-M supplements both exhibited higher levels of plasma IgM compared with the control group (*p* < 0.05).

### 3.5. Effect of AHAS on Serum Cytokine Levels in Lambs

As shown in Figure 5, the AHAS-H supplementation group had markedly higher concentrations of IL-4, IL-10, IL-17, IFN-γ, and TNF-α in plasma compared with the control group (*p* < 0.05). Additionally, both the AHAS-H and AHAS-M supplementation groups showed increased concentrations of IL-17 in plasma compared with the control group (*p* < 0.05).

### 3.6. AHAS Effects on Lamb Diversity and Abundance of Intestinal Microbiota

#### 3.6.1. Species Accumulation Curves and ASV/OTU

As shown in Figure 6A, the accumulation curve for gut microbiota showed the stability of species and suggested results sequencing captured the adequate diversity of species into the treatment groups for allowing subsequent analysis. The control group and the AHAS-H group contained 3988 and 4646 distinct operational taxonomic units (OTUs), respectively, and the number of intestinal microorganisms shared between the two groups was 777 (Figure 6B).

#### 3.6.2. Alpha and Beta (α and β) Diversity Analysis

Figure 7A shows the effect of AHAS on the intestinal tract microbacteria in lambs at α-diversity. The results of the continuum analysis for each group were deemed reliable and complete within a coverage rate of 0.997. The AHAS groups revealed higher values for the intestinal Shannon, Chao1, Simpson, Faith PD, Pielou E, and examined species indices in comparison to the control group. The AHAS group had significantly higher Faith PD indices than the control group (*p* < 0.05). Figure 7B displays the results of AHAS and UPGMA clustering analysis on the β-diversity of intestinal tract bacteria in lambs. The control group had obvious clustering, and some differences were recorded between the control and AHAS-H groups. The results of the PCoA analysis in Figure 7C show the principal components between AHAS-H and control group were found to be 23.5% of the total variation, and the distance is relatively close. The analysis of NMDS results shown in Figure 7D, with a strain value of 0.0524, showed that there were some variances in the composition and structure of the microbiota between the control and AHAS-H groups.

### 3.7. AHAS Impact on the Microbial Community Shape and Composition

The taxonomic composition of each group of lambs was analyzed further with reference to the specific variations in the microbial community. At the level of phylum (Figure 8A), the AHAS-H group showed a higher abundance of *Bacteroidetes*, *Firmicutes*, and *Actinobacteria* than the control group, while the abundance of *Proteobacteria* and *Spirochaetes* was comparatively lower. At the genus level (Figure 8B), the AHAS-H group showed an increase in the abundance of beneficial bacteria such as *Blautia*, *Prevotella*, and *Lactobacillus*, and a decrease in the abundance of the harmful bacterium *Moraxella*, but these differences were not statistically significant (*p* > 0.05). As indicated in Figure 8C, at the level of genus, there were five distinct communities with significant alterations (*p* < 0.05). The AHAS-H group showed the highest impact, with genera such as *Sphingomonas*, *Ralstonia*, and *Flavobacterium* (all with LDA scores [log10] > 3). AHAS raised the number and abundance of colonies, and the majority of bacteria were beneficial, which may contribute to promoting the host’s gut health. Metabolic pathway differential analysis (Figure 8D) indicated that the AHAS-H group significantly improved the activities of tetracycline biosynthesis, camphene and pinene degradation, caprolactam degradation, ketone bodies degradation and synthesis, tryptophan metabolism, degradation of lysine, metabolism of fatty acid, and metabolism of glycolipid, compared with the control group (*p* < 0.01).

### 3.8. Effect of AHAS on SCFAs in Lambs

The primary product of intestinal microbiota is SCFAs, which play a vital role in sustaining intestinal health. The results of the AHAS study on intestinal SCFAs in lambs are shown in Table 2. The range of valeric, acetic, propionic, and butyric acid was markedly increased in the AHAS-H group (*p* < 0.05) compared with the control group. These results of the current study reveal that AHAS-H effectively promotes the synthesis of various SCFAs.

## 4. Discussion

### 4.1. Growth Performance

Plant sugars have strong biocompatibility and low toxicity, degradability, and effective immune-stimulating effects [23,24]. Growth performance is an important health-based indicator of an animal’s growth at different stages. Body weight, height, chest circumference, feed intake, and daily weight gain are closely related to growth performance [25]. Weight change and the height index reflect the longitudinal development of the lamb’s skeleton, particularly the development of the limb bones [26]. The chest index, on the other hand, indicates more lateral development of the thorax and trunk of the lamb and is associated with the development of the ribs and thoracic vertebrae [26]. These indicators provide insight into the health and overall growth of lambs. In the current study, the accumulation of different doses of AHAS in the diet markedly improved the growth performance of lambs with dietary AHAS supplementation (Table 1). Lambs’ final body weight, average daily weight gain, final body height, final body slope length, final bust length, and final pipe circumference length were markedly raised in the AHAS-H group compared with the control group (*p* < 0.05). This suggests that AHAS-H positively affects the growth, development, and health of lambs, thereby improving their growth performance.

### 4.2. Antioxidant Status

Antioxidant enzymes play a key role in this process by defending against the damage caused by free radicals. Among these enzymes, T-AOC is a comprehensive indicator of the overall capacity of all serum antioxidants to combat oxidative damage [27]. GSH-PX is a hepatic protein essential for neutralizing hydroxyl radicals and peroxides produced during cellular respiration [27]. MDA is a secondary product of the process of lipid peroxidation and may be used to evaluate the extent of oxidative stress and lipid peroxidation [27,28]. T-SOD is an enzyme that catalyzes the asymmetric reaction of superoxide radicals and thus protects cells from damage by superoxide radicals [28]. Research indicates that plant sugars possess antioxidant properties and may effectively eliminate free radicals from the body, hence preserving oxidative balance [29]. Li et al. reported that dietary Artemisia ordosicacrude polysaccharides can significantly increase serum GSH-Px, T-SOD, and CAT activity in sheep [27]. Chen et al. suggested that dietary Lycium barbarum polysaccharides enhanced the serum GSH-Px, SOD, and CAT activities, in addition to reducing the MDA content [30]. The study results in Figure 3 reveal that the serum GSH-PX, T-AOC, and T-SOD contents of lambs in the AHAS-H group was markedly raised compared with the control group (*p* < 0.05), while the serum MDA content in the AHAS-H group was markedly lower than that of the control group (*p* < 0.05). These findings show that AHAS-H has the obvious ability to raise the action of antioxidant enzymes in the body, effectively scavenge free radicals produced during metabolic processes, maintain oxidative balance, and promote overall body health.

### 4.3. Immune Responses

The immunological system plays a key role in preserving animal health and facilitating growth via its ability to combat pathogens. Immunoglobulins play a vital role in the regulation of normal humoral immune response. The function of IgA as an antigen-specific binding is expressed in the mucosal area and protects the intestinal mucosa from infection [31]. IgG is mainly found in the blood and is expressed throughout the body, showing antiviral and antibacterial effects while also regulating the body’s immune system [32]. IgM plays a crucial role in the body fluid of adolescent animals as it boosts the immune cells’ ability to engulf harmful bacteria and combat pathogens [33]. As shown in Figure 4, serum IgG and IgA levels in the AHAS-H group were significantly higher than those in the control group (*p* < 0.05), and serum IgM levels in the AHAS-H and AHAS-M groups were significantly higher than those in the control group (*p* < 0.05). These findings suggest that AHAS-H effectively enhanced IgA and IgG secretion as well as serum IgM production, thereby boosting immune function within the intestines and throughout the body.

Cytokines such as (IL-4, IL-10, IL-17, TNF-α, and IFN-γ) play a key role in humoral and cellular immunity by raising production of lymphocytes and stimulating the peripheral immune system [34]. IL-4 and IL-10 are molecules that play a crucial role in maintaining immunological balance, minimizing tissue damage, decreasing inflammatory responses, and promoting antibody production [35]. IL-17 helps to enhance the barrier protective properties of the gut and mucosa and reduce the invasion of pathogens [36]. IFN-γ amplifies the immunological response of Th1 cells, promoting cellular immunity, cytotoxicity, and antibody class switching [37]. TNF-α plays a crucial role in controlling inflammation, eradicating infections, and repairing injured tissues [38]. As shown in Figure 5, the AHAS-H group exhibited substantially elevated levels of IL-4, IL-10, IL-17, IFN-γ, and TNF-α in their blood compared with the control group (*p* < 0.05). This study revealed that AHAS has the capability to influence cytokines linked with several subgroups of T helper cells. For instance, it affects TNF-α in Th0 cells, IFN-γ in Th1 cells, IL-4 in Th2 cells, IL-10 in Treg cells, and IL-17 in Th17 cells. Therefore, AHAS may effectively stimulate the production of different cytokines and antibodies in the body, enhancing tissue regeneration and aiding in the elimination of infections, reducing inflammation, and enhancing general immune function.

### 4.4. Effects of AHAS on Fecal Microbiota Dynamics

The complex intestinal microbiota plays a vital role in the food energy of nutrient absorption, assimilation, and digestion, which is closely linked to gut health and diseases. This study has shown that the addition of plant sugars to lamb diets enhances the probity of the intestinal barrier, enhances the immune response in the intestines, and increases the variety of microorganisms present [39]. The study examined the influence of AHAS on the gut microbiota and SCFAs in lambs using 16S rRNA sequencing methods. Species accumulation curves are often used to evaluate the abundance of sample sizes for assessing the diversity of a community, as well as to quantify and forecast the number of species in a given community. The species accumulation curves in this study approached a state of equilibrium as shown by sufficiently high sample sizes that allowed for the replication of the community’s species composition (Figure 6A). The study of microbial communities involved the use of ASV/OTUs. The current results (Figure 6B) suggest that the unique number of OTUs in the AHAS-H and control groups were 4646 and 3988, indicating the AHAS-H treatment group effectively increased the intestinal microbial diversity and community abundance in lambs. The α-diversity is a measure of species diversity within the intestinal microbial community. It has been found that data samples are reliable if the sample treatment exposure is greater than 97% [40]. The sample coverage in this study surpassed 98% (Figure 7A), indicating that the sequencing data accurately depicted the variety of species and composition of the lamb intestinal microbiota. The Faith PD index (*p* < 0.05) indicated that the AHAS-H group had a substantially increased abundance of intestinal bacteria in lambs compared with the control group (Figure 7A). The study demonstrated that the administration of AHAS significantly enhanced the diversity of gut bacteria in lambs. The β-diversity study used Venn diagrams to examine the shared and distinct microbiota across various populations. The clustering patterns of bacterial groups within these taxa were examined using UPGMA cluster analysis [41]. Principal coordinate analysis (PCoA) was used to measure the degree of dissimilarity across bacterial groups from different taxa [41]. In the UPGMA analysis (Figure 7B) and PCoA analysis (Figure 7C), there were some differences recorded between AHAS-H and the control group, and the two groups’ primary components explained 23.5% of the overall variation. NMDS is a method used in the field of environmental research. The goal was to preserve the original connections between study objects in a multidimensional setting while also identifying, examining, and classifying them in a lower-dimensional space. The microbiota compositions of the control and AHAS groups exhibited few differences, as shown by the strain value of 0.0524 in the NMDS analysis (Figure 7D). The current results suggest that the gut microbiota exhibited more diversity in the AHAS groups as compared to the control group. Consequently, it is logical to infer that AHAS can diversify the range of gut-microbial communities.

The gut microbiota metabolizes carbohydrates to produce monosaccharides and SCFAs, which are significant for sustaining the health of the gastrointestinal tract. The most prevalent forms of gut bacteria reported in many studies are *Firmicutes* and *Bacteroidetes* [42]. The *Firmicutes* and *Bacteroidetes* present in the large intestine metabolize complex carbohydrates to produce SCFAs such as acetic acid, propionic acid, and butyric acid, thereby promoting body growth and enhancing intestinal immunity [43]. *Actinomycetes* enhance the growth performance of their hosts by converting food into microbial biomass and fermentation products that may be ingested by them [44]. *Proteobacteria* have a marked influenced on the composition of gut microbacteria [45]. At phylum level, the AHAS-high group exhibited a higher abundance of *Firmicutes*, *Bacteroidetes*, and *Actinobacteria* as compared to the control group (Figure 8A), and lower *Proteobacteria* than in the control group. *Blautia* is a taxonomic classification within the phylum *Firmicutes*. It is often present in the intestines and feces of animals and has probiotic properties [46]. *Prevotella* is classified as a member of the *Bacteroidetes* phylum [46]. *Prevotella* is essential for the production of SCFAs and the decomposition of cellulose in the gastrointestinal tract [47]. *Lactobacillus* is a prevalent strain of lactic acid bacteria that protects the integrity of the intestinal lining by suppressing the proliferation of harmful bacteria, therefore reducing both inflammation and infection [48]. *Moraxella* is a pathogenic bacterium belonging to the *Proteobacteria* phylum and has the potential to induce respiratory infections, conjunctivitis, and several other illnesses [49]. Figure 8B shows that the AHAS-H group had a greater number of beneficial bacteria, such as *Blautia*, *Prevotella* and *Lactobacillus*, and reduced the abundance of the harmful bacteria *Moraxella* compared to the control group. Consequently, AHAS-H enhanced the intestinal microbiota composition in lambs, which had an important beneficial effect on growth performance, intestinal health, and overall health via the promotion of beneficial bacteria and reduction of detrimental bacteria.

Gut bacteria are the main internal producers of SCFAs due to their metabolic activity [50]. They are essential for maintaining organismic homeostasis. Butyric acid promotes proper immune system function by supplying energy to the cells that line the intestines [50]. Propionic acid hinders the dissemination of detrimental microorganisms and toxins via the mucosal folds of the intestines, preserving the well-being of the intestinal epithelium and fortifying the intestinal mucosal barrier [51]. Valeric acid is acknowledged as an energy substrate in the metabolic processes of gut bacteria [52]. The gut microbiota produces acetic and valeric acids, which are the most common short-chain fatty acids. Intestinal epithelial cells assimilate these substances, using them as an energy source and for their antibacterial properties [52]. The acids mentioned play a part in preserving the gut microbiota and also prevent the growth of harmful bacteria [53]. The AHAS-H group in this study had markedly raised the levels of acetic, butyric, valeric, and propionic acids compared with the control group (Table 2; *p* < 0.05). This demonstrates that AHAS-H treatment improved gastrointestinal health by increasing the production of SCFAs and modulating the sustainability and capability of the gut microbiota.

## 5. Conclusions

In summary, the incorporation of AHAS into the diet can increase the growth rate and intestinal immunological function of lambs. Some effects of AHAS include enhanced antioxidant enzymes, release of cytokines and antibodies, abundance of useful bacteria, decreased presence of dangerous bacteria, and augmented synthesis of SCFA. Furthermore, there is an enhancement in growth efficiency. AHAS is a naturally occurring substance which has a positive impact on animal health, and it can enhance the growth rate and the immunological functions of unweaned lambs.

## Figures and Tables

**Figure 1 animals-14-02402-f001:**
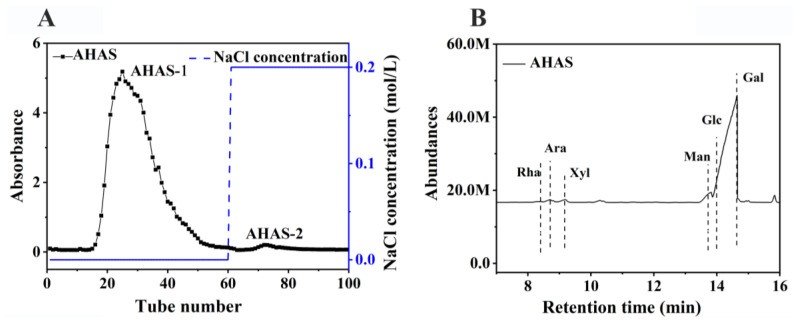
Composition and structural analysis of AHAS. (**A**) Elution curve of AHAS, (**B**) monosaccharide composition of AHAS.

**Figure 2 animals-14-02402-f002:**
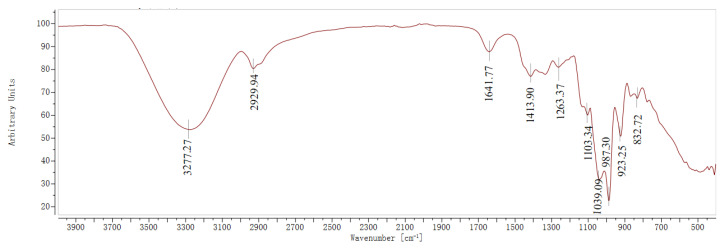
FT-IR analysis of AHAS.

**Figure 3 animals-14-02402-f003:**
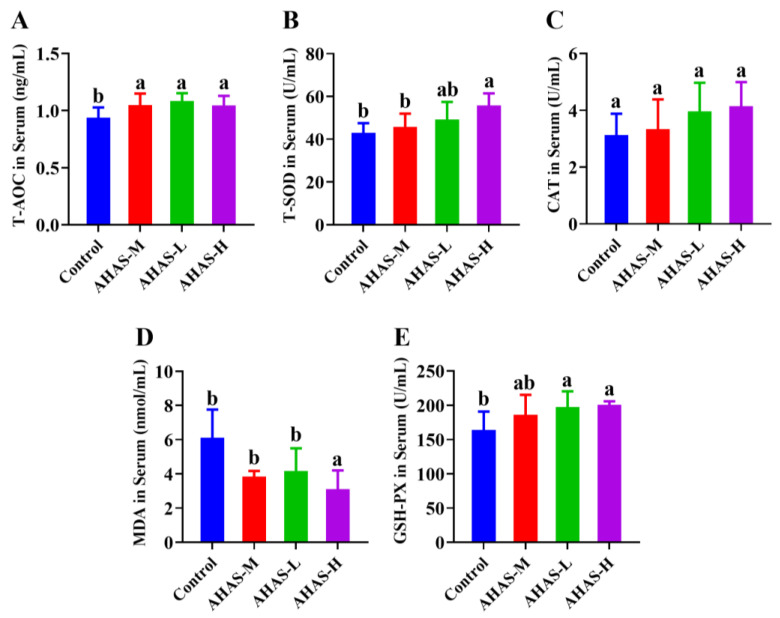
Effect of AHAS supplementation on serum antioxidant status of lambs (*n* = 7). (**A**) T-AOC, (**B**) T-SOD, (**C**) CAT, (**D**) MDA, (**E**) GSH-Px. Note: Bars that have distinct superscripts (a,b) exhibit significant differences (*p* < 0.05).

**Figure 4 animals-14-02402-f004:**
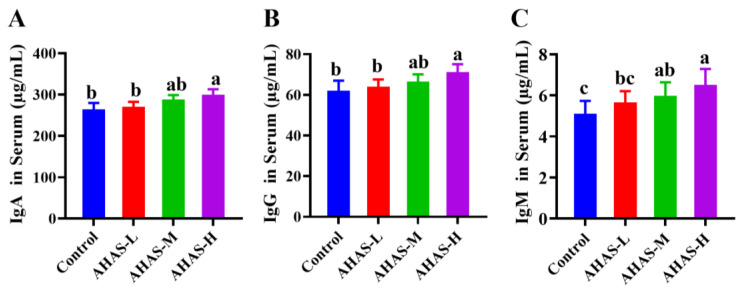
The impact of AHAS on plasma IgA, IgG, and IgM release in lambs (*n* = 7). The expression levels of IgA (**A**), IgG (**B**), and IgM (**C**) secretion into the lumen, as evidenced by ELISA assay. Note: Means within a row without the same superscripts (a–c) differ significantly (*p* < 0.05).

**Figure 5 animals-14-02402-f005:**
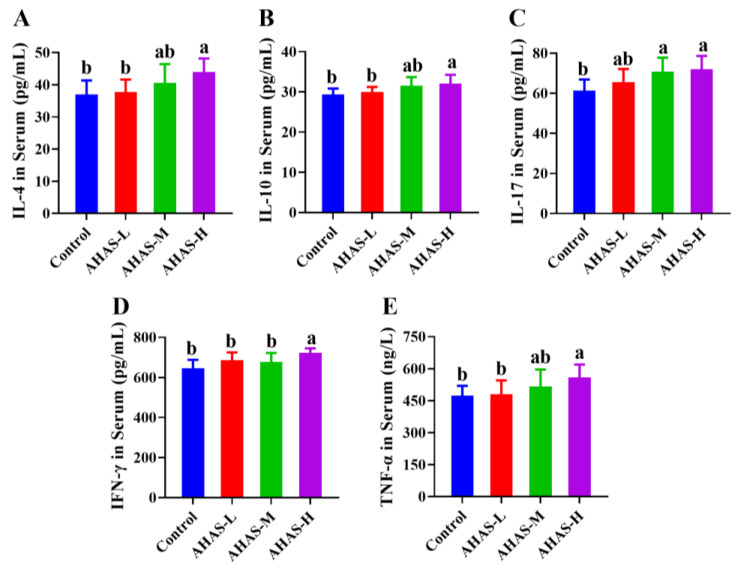
The impact of AHAS on plasma IL-4, IL-10, IL-17, TNF-α, and IFN-γ release in lamb (*n* = 7). The expression levels of (**A**) IL-4, (**B**) IL-10, (**C**) IL-17, (**D**) TNF-α, and (**E**) IFN-γ, as evidenced by ELISA. Note: Means within a row without the same superscripts (a, b) differ significantly (*p* < 0.05).

**Figure 6 animals-14-02402-f006:**
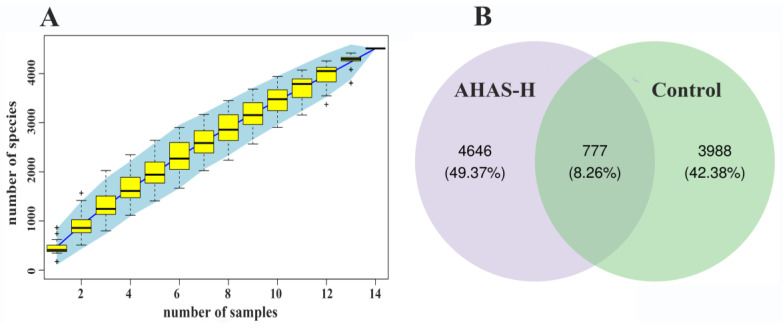
Species accumulation curves and ASV/OTU (*n* = 7). (**A**) Species accumulation curves, (**B**) comparative OTU analysis of the intestinal microbiota of lambs.

**Figure 7 animals-14-02402-f007:**
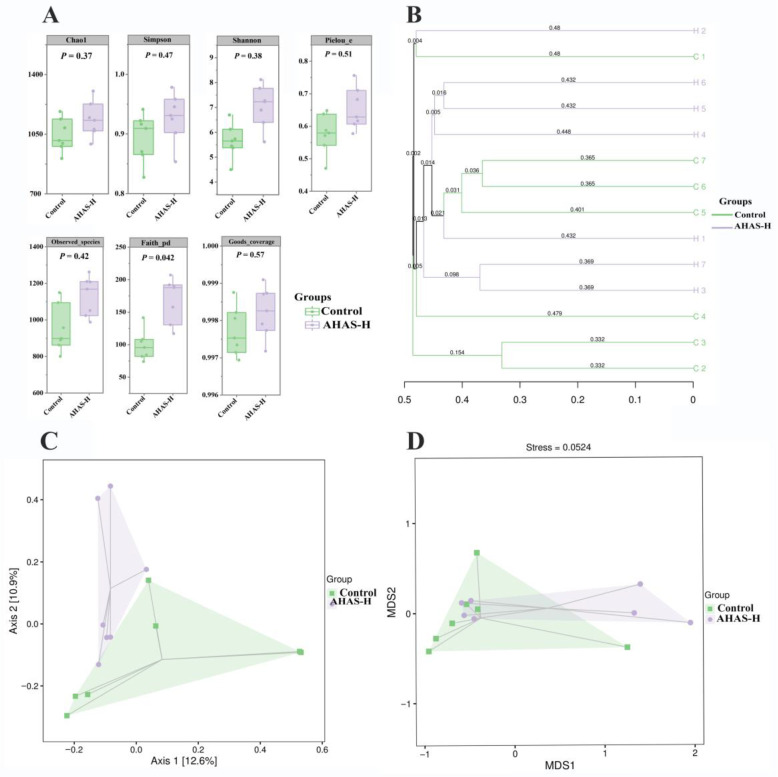
α-diversity and β-diversity of different groups of microorganisms (*n* = 7). (**A**) α-diversity indexes, (**B**) hierarchical cluster analysis based on weighted UniFrac distances, (**C**) principal coordinate analysis (PCoA) using the Bray–Curtis distance matrix, (**D**) non-metric multidimensional scaling (NMDS) analysis based on unweighted UniFrac distances.

**Figure 8 animals-14-02402-f008:**
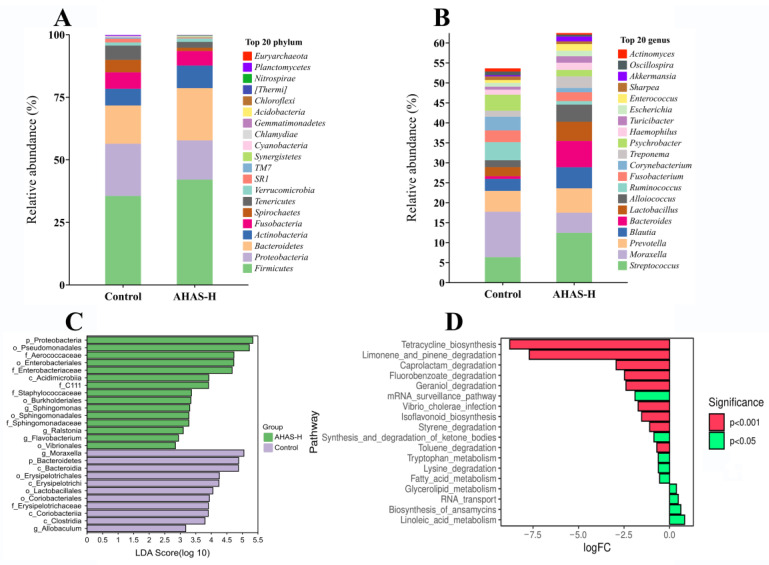
Microbiological analysis of AHAS-H group and control group (*n* = 7). Microbial composition profiles of two treatment groups at phylum (**A**) and genus (**B**) levels. (**C**) LDA effect size analysis (LEfSe). (**D**) Differential analysis of metabolic pathways: positive values of logFC on the horizontal axis represent up-regulation (log2 (fold change)) in group B relative to group A, while negative values indicate down-regulation. The vertical axis is labeled with different KEGG metabolic pathway names. The degree of significance is denoted by varying colors.

**Table 1 animals-14-02402-t001:** Effects of AHAS on growth performance and body size of lambs (*n* = 7).

Indicator	Control	AHAS-L	AHAS-M	AHAS-H
Initial body weight (kg)	2.15 ± 0.09	2.10 ± 0.27	2.07 ± 0.24	2.17 ± 0.36
Final body weight (kg)	4.54 ± 1.24 ^b^	4.77 ± 0.68 ^ab^	4.98 ± 0.62 ^ab^	5.57 ± 0.97 ^a^
Average daily gain (g/d)	120.31 ± 12.24 ^b^	133.28 ± 20.04 ^ab^	147.07 ± 15.15 ^ab^	170.8 ± 15.43 ^a^
Initial body height (cm)	28.69 ± 2.07	29.58 ± 1.41	28.90 ± 2.06	29.33 ± 1.46
Final body height (cm)	39.72 ± 1.28 ^c^	43.14 ± 1.70 ^bc^	45.97 ± 1.55 ^b^	48.59 ± 2.92 ^a^
Initial body slope length(cm)	28.63 ± 3.04	28.88 ± 1.28	28.83 ± 1.89	29.72 ± 1.34
Final body slope length(cm)	40.88 ± 3.15 ^c^	43.47 ± 2.70 ^bc^	42.22 ± 2.35 ^b^	46.31 ± 2.22 ^a^
Initial bust length(cm)	29.87 ± 1.83	29.24 ± 1.12	28.96 ± 1.53	29.91 ± 0.74
Final bust length (cm)	40.49 ± 2.93 ^c^	44.25 ± 2.44 ^b^	44.35 ± 1.69 ^b^	49.58 ± 3.40 ^a^
Initial pipe circumference length (cm)	3.38 ± 0.37	3.25 ± 0.286	3.34 ± .163	3.25 ± 0.19
Final pipe circumference length (cm)	4.32 ± 0.17	4.36 ± 0.19	4.68 ± 0.24	4.82 ± 0.17

Note: Means within a row without the same superscripts (a–c) differ significantly (*p* < 0.05).

**Table 2 animals-14-02402-t002:** Effect of AHAS on SCFAs in lambs (*n* = 7).

Items	Control	AHAS-H
Acetic acid (µmol/mL)	2320.04 ± 324.79 ^b^	3126.91 ± 312.47 ^a^
Butyric acid (µmol/mL)	325.12 ± 56.55 ^b^	517.71 ± 169.01 ^a^
Isovaleric acid (µmol/mL)	68.32 ± 15.53	84.06 ± 17.83
Valeric acid (µmol/mL)	57.92 ± 17.18 ^b^	106.82 ± 23.54 ^a^
Caproic acid (µmol/mL)	0.7235 ± 0.42	2.008 ± 0.58
Propionic acid (µmol/mL)	819.49 ± 69.51 ^b^	966.46 ± 82.58 ^a^
Isobutyric acid (µmol/mL)	76.71± 24.07	108.15 ± 12.52

Note: Means within a row without the same superscripts (a, b) differ significantly (*p* < 0.05).

## Data Availability

The data presented in this study are available on request from the corresponding author.

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
