# Peer review of "Regulatory Effects of Alhagi Honey Small-Molecule Sugars on Growth Performance and Intestinal Microbiota of Lambs"

_animals, 2024, doi:10.3390/ani14162402_

Round 1

Reviewer 1 Report

Comments and Suggestions for Authors

Dear Authors!

Based on your manuscript, AHAS can be a "miracle weapon" in feeding ruminants. Would it really be? How much did it cost to feed AHAS for 28 days? How much does it cost if we have 500 ewes? Or what if there are 2000 or 3000?

Basically, I think there are too many points in the manuscript. It would be better to focus on only two or three of these (e.g. the effect of AHAS on growth performance and appetite of lambs, or something similar).

All of my other comments and recommendations can be found in the attached version.

best regards

Comments on the Quality of English Language

Author Response

Reviewer 1

  1. Line7 Please correct the name of the institute.

Answer:The correct name of the institute is “Xinjiang Key Laboratory of New Drug Study and Creation for Herbivorous”

  1. Line18 It is not needed. (same as in row 28)

Answer: It has been changed in revised manuscript.

  1. Line20 Please define it at first use. What groups did you create and what were the differences between the groups?

Answer: It has been changed in revised manuscript.

  1. Line39 In recent years, China's lamb production has been shifted towards large-scale monoculture-based industrial farming in response to increasing consumer demand for meat.

Change to  In recent years, China's lamb production has been shifted towards large-scale monoculture-based industrial farming in response to increasing consumers’ demand for meat.

Answer: It has been changed in revised manuscript.

Please refine the keeping method of the animals, becouse all of these details can be effect on the results:

- what is the keeping system (extensive - natural way; or intensive - artificial keeping);

- how do you keep the lambs (under dams or separated to in artificial keeping system), when do you separate them from dams

- how do you feed the lambs (besed on dams' milk - natural way; or based on milk powder - artificial way);

- did the lambs get any solid feed from the 2nd-3rd week of life, if yes, what was that and what is its composition;

- what medical treatments did the lambs get during the trial;

- etc....

Answer: It has been changed in revised manuscript.

What was composition of the "normal diet"?

Answer: "normal diet" is normal saline

Please correct this sentence.

Answer: It has been changed in revised manuscript.

Line131 This image is not required, it has no additional information. The criteria for grouping are clearly defined in the text. It is recommended to delete it.

Answer: It has been deleted in revised manuscript.

Line139  Do you mean: "before the feeding time of the following day"?

Answer: “blood samples were collected in 10 mL vacutainer vials before the next day of feeding the animal diet” change to “blood samples were collected in 10 mL vacutainer vials before morning feeding”

line151  After study trial completion, collect the fecal samples from each group of lambs (n=7, per group)  change to  After study trial completion, fecal samples were collected from each group of lambs (n=7, per group)

Answer: It has been changed in revised manuscript.

line181  “formed of Man, Ara, Rha, Xyl, Glc, and Gal,” Please define them at first use in the text.

Answer: formed of mannose (Man), arabinose (Ara), rhamnose (Rha), xylopentaose (Xyl), glucose (Glc), and galactose (Gal)

The differences are very clear, but I'm not sure if it can come only by the feeding of minimal AHAS dose. Growht performance is effected by lot of things as genetical background, mother's effects, feeding system (we have no infos about it), keeping system etc.

Answer:The relevant background is added to the "2.4. Animal experimental design" in revised manuscript.

At the same time, AHAS can increase the appetite of animals. Do you have information on how the feed intake of the lambs varied in each group during the trial?

Answer: During the trial, lambs fed lamb pellet feed freely, and feed intake was not monitored. Thanks to the reviewers for reminding, we will focus on this issue in the following experiments.

Did you check the condition scores of the lambs, as well? This is one of the easiest ways to evaluate the actual physiological status of the animals.

Answer: We did not keep a detailed record of the condition scores of the lambs. At the beginning of the experiment, we selected lambs with condition scores of about 3 for the experiment "A total of 28 Hu lambs, with each lamb weighing 2.13 ± 0.12 kg and being around 7-days old were randomly divided into 4 groups (n=7)." After the experiment, the condition scores of lambs in AHAS group was about 4.

On the other hand, meat quality tests should be also beneficial to find out different meat quality indicators as meat-to-fat ratio or fat content. Because, growth permormance is not equal to fattening...

Answer: Meat quality testing is the key to finding different meat quality indicators. Considering that the object of this experiment was lambs rather than commercial meat sheep for slaughter, and the purpose of this experiment was to detect the effects of AHAS on the growth, immune function and intestinal flora of lambs, the meat quality was not tested.

Is it economically beneficial for farmers to feed AHAS to lambs? What is the price of AHAS on the market and how can farmers get it?

Answer: Direct data on AHAS production are not available. However, we know that AHAS is the sugar secreted by Alhagi pseudalhagi, and the sugar produced by Alhagi pseudalhagi is about 45000-150000 kg/square kilometers. The distribution area of Alhagi pseudalhagi in Xinjiang is about 173,000 square kilometers, and the larger area and production of Alhagi pseudalhagi are distributed in Afghanistan, Azerbaijan, Pakistan, Kazakhstan and other East Asian countries[1, 2]. The retail price of Alhagi honey is about 80 yuan /kg and the extraction rate of AHAS is 76.56 %, so the price of AHAS is about 106 RMB /kg. The dosage of AHAS-H group lambs is 800 mg/kg. This experiment is used on lambs between 2 and 4 kg. Therefore, the cost of feeding AHAS to one sheep at a time is 0.17 of AHAS -0.51 of AHAS (component price).

References

[1]Ahmad N, BibiY, Saboon, et al. Traditional uses and pharmacological properties of Alhagi maurorum: A review[J]. Asian Pacific Journal of Tropical Disease. 2015; 5(11): 856-861

[2]Hui J. The comprehensive evaluation of feed value of 4lhagisparsifolia Shap and alfalfa mix-silage[D]. China Agriculture University. 2017;11:1-14. (in Chinese)

Reviewer 2 Report

Comments and Suggestions for Authors

Thank You for the submitted paper. I have a few comments on it:

Line 18, there is no need to add a word "sheep" before "lambs" (the same in line 26); also because a lot of people read only abstract it is good to explain what AHAS-H group mean

Line 23, the latin names should be written in italics (the same in lines 57,60,61,71,75, 283- 287,290,291,424,425,429-432,434-436,438,439,441,442)

Lines 58-59, in my opinion this sentence doesn't fit to the introduction

Line 61, there is no need to add "Desv"

Line 72, the abbreviation needs an explanation

Lines 78, 79 instead of "are", it is better to use "were". You can also read carrefully the rest of paper to correct such mistakes

Lines 87 and 122, first You mentioned that there were 40 lambs, next 28. It needs to be clarify. Also what was the sex of lambs?

Lines 122-129, was the dosage  increase during the trial? What was the method of administration?

Line 144 "T-AOC" instead of "TAOC"

Line 151 "the fecal samples were collected"

Line 212 what is "bosom length"If You mean "bust length", it is good to use one form

Line 217 it is good to put the description under every table or figure (the same for line 228)

Lines 218, 229 delete

Line 327 "final chest girth" is doubled. Also "final chest girth" and "final body diagonal length" aren't mentioned before

References, if You are using abbreviations for the journals, please make sure, that You use for everyone, line 497

Author Response

Reviewer 2

Thank You for the submitted paper. I have a few comments on it:

Line 18, there is no need to add a word "sheep" before "lambs" (the same in line 26); also because a lot of people read only abstract it is good to explain what AHAS-H group mean

Answer: It has been changed in revised manuscript.

Line 23, the latin names should be written in italics (the same in lines 57,60,61,71,75, 283- 287,290,291,424,425,429-432,434-436,438,439,441,442)

Answer: It has been changed in revised manuscript.

Lines 58-59, in my opinion this sentence doesn't fit to the introduction

Answer: This sentence was deleted in the revised manuscript.

Line 61, there is no need to add "Desv"

Answer: It has been changed in revised manuscript.

Line 72, the abbreviation needs an explanation

Answer: The explanation of SCFAs is given in Line 54

Lines 78, 79 instead of "are", it is better to use "were". You can also read carrefully the rest of paper to correct such mistakes

Answer: We have examined the full text of the manuscript and similar errors have been changed in revised manuscript.

Lines 87 and 122, first You mentioned that there were 40 lambs, next 28. It needs to be clarify. Also what was the sex of lambs?

Answer: The 40 lambs in Lines 87 were writing errors. Correct are 28 lambs, 7 per group. We modified the number in the Lines 87 and clear the sex of the lamb.

Lines 122-129, was the dosage increase during the trial? What was the method of administration?

Answer: It has been changed in revised manuscript.

Line 144 "T-AOC" instead of "TAOC"

Answer: It has been changed in revised manuscript.

Line 151 "the fecal samples were collected"

Answer: It has been changed in revised manuscript.

Line 212 what is "bosom length"If You mean "bust length", it is good to use one form

Answer: Final bust length. It has been changed in revised manuscript.

Line 217 it is good to put the description under every table or figure (the same for line 228)

Answer: It has been changed in revised manuscript.

Lines 218, 229 delete

Answer: It has been changed in revised manuscript.

Line 327 "final chest girth" is doubled. Also "final chest girth" and "final body diagonal length" aren't mentioned before

Answer: It has been changed in revised manuscript.

References, if You are using abbreviations for the journals, please make sure, that You use for everyone, line 497

Answer: It has been changed in revised manuscript.

Reviewer 3 Report

Comments and Suggestions for Authors

The manuscript is interesting and the study design is sound.

I have some suggestions as follows below:

L60: I am wondering the amount of Alhagi honey production in the world and in China?

L79: Please define AHAS here as you have not defined earlier in the main manuscript.

L94-98: I am unclear here what are meant by macromolecules and small molecules? Please explain in a better way here.

Figure 1: I think this figure is not needed.

L162: A short description of bioinformatics should be provided here to be independent.

Table 1: Would you please provide feed intake in different groups to understand if the growth effect was due to feed intake or other reasons.

Figure 8: The quality may be improved as letter font size is small to read.

Table 2: TVFA concentration should also be provided.

Discussion

L315: what was the reason of increased body weight gain?

I am wondering the AHAS molecules were not degraded in the rumen by ruminal microorganisms?

Immune response: AHAS molecules increased both inflammatory and anti-inflammatory responses as evident from cytokines concentration. Why is it so?

Comments on the Quality of English Language

Acceptable

Author Response

Reviewer 3

The manuscript is interesting and the study design is sound.

I have some suggestions as follows below:

L60: I am wondering the amount of Alhagi honey production in the world and in China?

Answer: Direct data on AHAS production are not available. However, we know that AHAS is the sugar secreted by Alhagi pseudalhagi, and the sugar produced by Alhagi pseudalhagi is about 45000-150000 kg/square kilometers. The distribution area of Alhagi pseudalhagi in Xinjiang is about 173,000 square kilometers, and the larger area and production of Alhagi pseudalhagi are distributed in Afghanistan, Azerbaijan, Pakistan, Kazakhstan and other East Asian countries[1, 2].

References

[1]Ahmad N, BibiY, Saboon, et al. Traditional uses and pharmacological properties of Alhagi maurorum: A review[J]. Asian Pacific Journal of Tropical Disease. 2015; 5(11): 856-861

[2]Hui J. The comprehensive evaluation of feed value of 4lhagisparsifolia Shap and alfalfa mix-silage[D]. China Agriculture University. 2017;11:1-14. (in Chinese)

L79: Please define AHAS here as you have not defined earlier in the main manuscript.

Answer: It has been changed in revised manuscript.

L94-98: I am unclear here what are meant by macromolecules and small molecules? Please explain in a better way here.

Answer: Small molecular sugars include monosaccharides and oligosaccharides (molecular weight between 200 and 3000 Da), and macromolecular t-sugars usually refer to sugars with a molecular weight of more than 3000 Da. Short explanations have been added to the revised manuscript

Figure 1: I think this figure is not needed.

Answer: It has been deleted in revised manuscript.

L162: A short description of bioinformatics should be provided here to be independent.

Answer: It has been deleted in revised manuscript.

Table 1: Would you please provide feed intake in different groups to understand if the growth effect was due to feed intake or other reasons.

Answer: During the trial, lambs fed lamb pellet feed freely, and feed intake was not monitored. Thanks to the reviewers for reminding, we will focus on this issue in the next experiments.

Figure 8: The quality may be improved as letter font size is small to read.

Answer: It has been deleted in revised manuscript.

Table 2: TVFA concentration should also be provided.

Answer: We tested the main short-chain fatty acids, but not the total volatile fatty acids (TVFA). Thanks to the reviewer for reminding us that we will pay attention to this issue in the next experiment.

L315: what was the reason of increased body weight gain?

Answer: The improvement of growth performance (body weight gain) is closely related to the effect of AHAS on enhancing the abundance of beneficial bacteria (Bacteroidetes, Lactobacillus sp, et al), reducing the abundance of harmful bacteria and promoting a variety of short-chain fatty acids. The relevant content has been added in revised manuscript in Line 446-471.

I am wondering the AHAS molecules were not degraded in the rumen by ruminal microorganisms?

Answer: Studies have shown that sugars cannot be directly absorbed and utilized by the body (except for monosaccharides). Sugars need to be absorbed by microorganisms in the gastrointestinal tract and decomposed into small molecules for absorption and utilization by the body. At the same time, these sugars can regulate the gastrointestinal microbe[1-3]. We have no direct evidence on whether AHAS is degraded by rumen microorganisms, but according to the changes of intestinal flora, we assume that it must be absorbed by gastrointestinal flora. Thank you very much for the editor's question, we will focus on this problem in the next experiment.

References

[1] Song W, Wanag Y, Li gong. Modulating the gut microbiota is involved in the effect of low-molecular-weight Glycyrrhiza polysaccharide on immune function[J].Gut Microbes, 2023, 15(2):1-14.

[2] Xie S Z, Liu B, Ye H Y, et al. Dendrobium huoshanense polysaccharide regionally regulates intestinal mucosal barrier function and intestinal microbiota in mice[J]. Carbohydrate Polymers, 2019, 206: 149-162.

[3] Sha Z, Shang H, Miao Y, et al. Polysaccharides from Pinus massoniana pollen improve intestinal mucosal immunity in chickens[J]. Poultry Science, 2021, 100: 507-516.

Immune response: AHAS molecules increased both inflammatory and anti-inflammatory responses as evident from cytokines concentration. Why is it so?

Answer: A large number of studies have shown that sugar can improve the immune function by promoting the concentration of cytokines in the body, and the significant increase of cytokines in a certain range is the manifestation of the enhancement of immune function, rather than the manifestation of inflammation[1-3]. Only when cytokines are increased several times or tens of times compared to the blank control group, the inflammatory storm indicates that there is an inflammatory response in the body[4, 5]. It is obvious that the AHAS-H group can significantly promote the expression of various cytokines compared with the control group, but it has not reached the degree of inflammatory response.

References

[1] Wang J, Ge B, Li Z, et al. Structural analysis and immunoregulation activity comparison of five polysaccharides from Angelica sinensis[J]. Carbohydrate Polymers, 2016, 140: 6-12.

[2] Wen L, Sheng Z, Wang J, et al. Structure of water-soluble polysaccharides in spore of Ganoderma lucidum and their anti-inflammatory activity[J]. Food Chemistry, 2022, 373: 131374.

[3] Fu Y-P, Feng B, Zhu Z-K, et al. The Polysaccharides from Codonopsis pilosula Modulates the Immunity and Intestinal Microbiota of Cyclophosphamide-Treated Immunosuppressed Mice [J], Molecules, 2018, 23(7): 1801.

[4] Feng Z, Peng S, Wu Z, et al. Ramulus mori polysaccharide-loaded PLGA nanoparticles and their anti-inflammatory effects in vivo [J]. International Journal of Biological Macromolecules, 2021, 1(182):2024-2036.

[5] Feng Z, Jiao L, Wu Z, at al. A Novel Nanomedicine Ameliorates Acute Inflammatory Bowel Disease by Regulating Macrophages and T-Cells [J]. Molecular pharmaceutics, 2021 6;18(9):3484-3495.

Round 2

Reviewer 1 Report

Comments and Suggestions for Authors

All the reviewer's comments and recommendations have been accepted.

Author Response

Reviewer 1  Round 2

Based on your manuscript, AHAS can be a "miracle weapon" in feeding ruminants. Would it really be? How much did it cost to feed AHAS for 28 days? How much does it cost if we have 500 ewes? Or what if there are 2000 or 3000?

The results showed that AHAS had various effects on growth promotion, immune enhancement and intestinal microorganism regulation of lambs. The raw material price of Alhagi honey is more expensive than that of other medicinal plants, but the content of AHAS in Alhagi honey is very high, so the price of AHAS is relatively low. The retail price of Alhagi honey is about 80 RMB/kg and the extraction rate of AHAS is 76.56 %, so the price of AHAS is about 106 RMB /kg. The dosage of AHAS-H group lambs is 800 mg/kg. This experiment is used on lambs between 2 and 4 kg. Therefore, the cost of feeding AHAS to one sheep at a time is 0.17 RMB-0.51 RMB (Raw material price, not including the extraction reagent and labor costs). The cumulative use cost of a lamb on 28 days is approximately 10RMB

Basically, I think there are too many points in the manuscript. It would be better to focus on only two or three of these (e.g. the effect of AHAS on growth performance and appetite of lambs, or something similar).

Answer: This manuscript is a preliminary study on the effects of AHAS on the functions of lambs. The design of the experiment was uncertain as to which indicators AHAS could positively affect lambs. Therefore, there is no targeted in-depth study on 1-2 points. Thank you very much for the comments of the reviewers. Next, we will conduct in-depth studies on the growth-promoting and immune-enhancing effects of AHAS respectively.

All of my other comments and recommendations can be found in the attached version.

best regards

Dear reviewer, we have revised and answered your question one by one in Authors' Responses to Reviewer's Comments, and the revised part is marked in red in revised manuscript.

Reviewer 2 Report

Comments and Suggestions for Authors

Thank You for the resubmitted version of your paper. I have a few coments on it:

line 7 Shouldn't the "animals" in Xinjiang Key Laboratory of New Drug Study and Creation for Herbivorous animals be written in capital letter?

line 131, why after "with" the rest of the sentence is in brackets?

lines 131-132, I suggest the improvement of this sentence "... and the treatment group which was divided..."

line 138 there is no need to add "sheep" before lambs

line 147 I suggest the improvement of this sentence "...all the measurements were taken, such as body weight,..."

Author Response

Reviewer 2  Round 2

Thank You for the resubmitted version of your paper. I have a few coments on it:

line 7 Shouldn't the "animals" in Xinjiang Key Laboratory of New Drug Study and Creation for Herbivorous animals be written in capital letter?

Answer: It has been changed in revised manuscript.

line 131, why after "with" the rest of the sentence is in brackets?

Answer: It has been changed in revised manuscript.

lines 131-132, I suggest the improvement of this sentence "... and the treatment group which was divided..."

Answer: It has been changed in revised manuscript.

line 138 there is no need to add "sheep" before lambs

Answer: It has been changed in revised manuscript.

line 147 I suggest the improvement of this sentence "...all the measurements were taken, such as body weight,..."

Answer: It has been changed in revised manuscript.